# Neutrophil-to-Lymphocyte Ratios and Infections after Traumatic Brain Injury: Associations with Hospital Resource Utilization and Long-Term Outcome

**DOI:** 10.3390/jcm10194365

**Published:** 2021-09-24

**Authors:** Marina Levochkina, Leah McQuillan, Nabil Awan, David Barton, John Maczuzak, Claudia Bianchine, Shannon Trombley, Emma Kotes, Joshua Wiener, Audrey Wagner, Jason Calcagno, Andrew Maza, Ryan Nierstedt, Stephanie Ferimer, Amy Wagner

**Affiliations:** 1Department of Physical Medicine & Rehabilitation, University of Pittsburgh, Pittsburgh, PA 15213, USA; msl66@pitt.edu (M.L.); LEV23@pitt.edu (L.M.); naa86@pitt.edu (N.A.); JMM412@pitt.edu (J.M.); CLB227@pitt.edu (C.B.); tuo21498@temple.edu (S.T.); erk74@pitt.edu (E.K.); JMW264@pitt.edu (J.W.); audreywa@buffalo.edu (A.W.); jasoncalcagno12@gmail.com (J.C.); andrewjmaza@pitt.edu (A.M.); rnierstedt@oakland.edu (R.N.); 2Department of Infectious Diseases & Microbiology, University of Pittsburgh, Pittsburgh, PA 15213, USA; 3Department of Biostatistics, University of Pittsburgh, Pittsburgh, PA 15213, USA; 4Department of Emergency Medicine, University of Pittsburgh, Pittsburgh, PA 15213, USA; bartond2@upmc.edu; 5Division of Pediatric Rehabilitation Medicine, Department of Orthopaedics, West Virginia University, Morgantown, WV 26506, USA; slf587@gmail.com; 6Safar Center for Resuscitation Research, University of Pittsburgh, Pittsburgh, PA 15213, USA; 7Department of Neuroscience, University of Pittsburgh, Pittsburgh, PA 15213, USA; 8Center for Neuroscience, University of Pittsburgh, Pittsburgh, PA 15213, USA; 9Clinical and Translational Science Institute, University of Pittsburgh, Pittsburgh, PA 15213, USA

**Keywords:** neutrophil-to-lymphocyte ratio, traumatic brain injury, hospital acquired infection, hospital resource utilization, inflammation, mechanical ventilation, Glasgow outcome scale

## Abstract

Traumatic brain injury (TBI) induces immune dysfunction that can be captured clinically by an increase in the neutrophil-to-lymphocyte ratio (NLR). However, few studies have characterized the temporal dynamics of NLR post-TBI and its relationship with hospital-acquired infections (HAI), resource utilization, or outcome. We assessed NLR and HAI over the first 21 days post-injury in adults with moderate-to-severe TBI (*n* = 196) using group-based trajectory (TRAJ), changepoint, and mixed-effects multivariable regression analysis to characterize temporal dynamics. We identified two groups with unique NLR profiles: a high (*n* = 67) versus a low (*n* = 129) TRAJ group. High NLR TRAJ had higher rates (76.12% vs. 55.04%, *p* = 0.004) and earlier time to infection (*p* = 0.003). In changepoint-derived day 0–5 and 6–20 epochs, low lymphocyte TRAJ, early in recovery, resulted in more frequent HAIs (*p* = 0.042), subsequently increasing later NLR levels (*p* ≤ 0.0001). Both high NLR TRAJ and HAIs increased hospital length of stay (LOS) and days on ventilation (*p* ≤ 0.05 all), while only high NLR TRAJ significantly increased odds of unfavorable six-month outcome as measured by the Glasgow Outcome Scale (GOS) (*p* = 0.046) in multivariable regression. These findings provide insight into the temporal dynamics and interrelatedness of immune factors which collectively impact susceptibility to infection and greater hospital resource utilization, as well as influence recovery.

## 1. Introduction

Traumatic brain injury (TBI) is a common insult caused by an external force, often from automobile accidents, athletic injuries, falls, or violence. Approximately 1.7 million TBIs occur in the United States (US) annually [1], and they can be financially burdensome, as individuals with moderate-to-severe TBI typically incur, on average, between $8000 to over $33,000 in hospital charges solely during their acute care hospitalization [2]. TBI carries public health implications as a leading cause of morbidity and mortality, and disability often extends into the chronic phase of recovery, particularly among those with moderate-to-severe injuries [3,4]. Unfavorable outcomes after TBI may be attributable not only to the brain injury itself but also to the many secondary conditions that arise during hospitalization, such as non-neurologic organ dysfunction (NNOD), dysfunctional immune responses, and concomitant infections [5,6,7,8].

In the days following TBI, hospital-acquired infections (HAI) can have deleterious effects during acute care. HAIs often lead to longer hospital and intensive care unit (ICU) stays, higher probabilities of hospital readmission, higher mortality rates both during and after the initial acute hospitalization [7,8,9,10], and worse long-term outcomes [11,12,13]. Broad-spectrum prophylactic antibiotic use after TBI has low efficacy and contributes to the increasing threat of antibiotic-resistant bacteria [14]. Therefore, strategies to assess patient susceptibility to infections after TBI may reduce hospital cost burden, improve infection control and treatment, and reduce morbidity and mortality rates in this population [15].

HAI is both prevalent and deleterious, particularly among patients with TBI, and is an emerging consideration when delineating TBI outcomes [10]. Several studies have shown that acquiring concomitant infections is associated with poorer outcomes in critically ill patients with and without TBI [11,12,13,16]. Among patients who survived at least one year post-TBI, 19% of subsequent mortality was attributable to infectious causes [17], suggesting long-term derangements in immune function. This is consistent with recent experimental TBI studies demonstrating long-standing dysregulated immune function post-injury [18]. Hospital-acquired pneumonia (HAP) is a common HAI occurring among patients with TBI [11,19,20]. Increased susceptibility to HAI is frequently attributed to the increased incidence of mechanical ventilation [19], which ultimately leads to a risk of developing ventilator-associated pneumonia (VAP) [21,22]. Consequently, acute care infections coincide with longer mechanical ventilation, longer hospital lengths of stay (LOS) [22], increased risk for other complications such as NNOD [7,8,23], and increased risk for disability and death [5,11,20]. Individuals with acute TBI may be more susceptible to nosocomial infections compared to other ICU patients due to the degree of immunosuppression, need for invasive procedures, ventilation and urinary catheters, and co-occurring peripheral injuries [10], with reported infection rates up to 50% in this population [5,24,25,26]. Findings from experimental models support this trend of increased risk, with infected mice subjected to experimental TBIs having poorer recoveries and being more likely to die compared to infected sham animals [18]. Together, this literature demonstrates that infections post-TBI are common, impact recovery course, and are vital to understanding the complex mechanisms contributing to TBI global outcomes.

Recently, the Neutrophil-to-Lymphocyte Ratio (NLR) has been proposed as a rapid, easily obtainable, and objective metric having prognostic value in several populations with non-neurological disease and/or organ dysfunction. For example, NLR is reportedly an independent prognostic indicator of survival in small-cell lung cancer [27], gallbladder cancer [28], and hepatocellular carcinoma [29]. More recently, NLR is reported to be a prognostic indicator for acute respiratory infections such as COVID-19 [30,31]. In other facets of critical care, NLR may be a better early indicator of bacteremia in patients admitted to the emergency department than conventional tests such as white blood cell count, c-reactive protein, or neutrophil and lymphocyte counts alone [32]. 

Higher NLR reportedly also portends worse outcomes in central nervous system (CNS) pathologies, including glioblastoma [33] and ischemic stroke patients receiving mechanical thrombectomy [34]. More recently, NLR has emerged as a prognostic marker in the TBI population [35]. Retrospective studies demonstrate higher NLR (at admission and peak level) is associated with unfavorable outcomes and mortality after severe TBI [36,37,38]. There is, however, a gap in understanding how NLR relates to infection status and subsequent outcomes in patients with moderate-to-severe TBI. Using this easily calculated metric may support efficient preemptive approaches to addressing HAI post-TBI while reducing hospital resource utilization and affecting outcome.

We hypothesized that groups with higher NLR would be more susceptible to acquiring infections during acute care hospitalization and that infection would influence NLR status. To that end, we explored bidirectional associations between high NLR and infections and their coupled role in increasing hospital utilization, including days on mechanical ventilation and hospital LOS. We also explored how NLR and infection status, independently and collectively, influence global outcome six months post-injury. 

## 2. Methods

### 2.1. Study Design and Cohort Descriptions

The University of Pittsburgh Institutional Review Board (IRB) approved this prospective cohort study, which is a part of a larger study evaluating biological and clinical markers associated with TBI outcome. Informed consent was provided by participants or their next-of-kin when appropriate. The inclusion criteria for the larger study were age ≥16 years with a moderate-to-severe TBI as defined by a GCS score of <13 or a GCS score of 13–15 with sufficient medical documentation of moderate-to-severe injury (i.e., positive anatomic neuroimaging findings or focal neurologic signs) on day of injury.

Individuals were excluded from this study if they presented with any of the following criteria: penetrating head injury, prolonged cardiac or respiratory arrest at injury (occurring >30 min prior to admission), evidence of brain death within the first three days after injury, Abbreviated Injury Scale (AIS) score of five in regions other than the head or neck, untreated endocrine diseases, autoimmune disorders, ongoing neurological or neurodegenerative diseases, or history of cancer (malignant neoplasms) or concurrent cancer reported via ICD-9 codes upon acute care discharge. We further aligned our exclusion criteria for this specific analysis with Chen et al., a previous study evaluating NLR in severe TBI [37], wherein patients with a history of cardiac surgery or disorders such as coronary artery disease, congenital cardiac disease, congestive heart failure, and/or myocardial dysfunction were excluded due to literature suggesting strong NLR mortality associations among those with cardiac disorders [39]. Additionally, previous neurologic disorders and trauma such as history of stroke, previous moderate-to-severe head trauma, or seizure disorders were also excluded so as to not confound the current injury profile assessment and results interpretation.

Figure 1 summarizes data availability in a study cohort selection diagram. Additionally, this study specifically required the availability of at least two NLR data-points and infection data (*n* = 196) over the first three weeks post-injury, available through the University of Pittsburgh Medical Center (UPMC) system’s electronic medical records (EMR). For analysis purposes, we used the full cohort (*n* = 196) of individuals, that included a subset who died within 21 days of the injury, and an acute-care-survivors cohort (*n* = 175), which excluded those who died within 21 days of injury. The Glasgow Outcome Scale (GOS) was required for outcome analyses, which was obtained through follow-up participant interviews six months post-injury.

### 2.2. Demographic and Clinical Data Abstraction 

Demographic and clinical variables were collected via a combination of personal interview, the UPMC local trauma registry, and EMR review. These variables included: age, sex, race, mechanism of injury (MOI), best GCS score during the first 24 h post-injury, Injury Severity Scale (ISS) score, acute care hospital LOS, number of days of mechanical ventilation, and acute care computerized tomography (CT) injury findings.

Pre-specified brain lesion types were abstracted from CT reports and categorized. Intra-axial hemorrhages included intra-ventricular hemorrhage (IVH), intra-parenchymal hemorrhage (IPH), and contusions. Extra-axial hemorrhages included subdural hemorrhage (SDH), subarachnoid hemorrhage (SAH), and epidural hemorrhage (EDH). Diffuse axonal injury (DAI) and midline shift were also reported. A CT burden score, similar to neurological burden scores presented in our other work [40,41], was derived for each individual as an ordered sum of injury types present on serial CT reports over the duration of the acute care hospital stay that were obtained as a part of clinical care. Incidence of the following injury types were summed to give a total CT burden score: IVH, IPH, contusions, SDH, SAH, EDH, and DAI.

Acute care measures, including microbiology culture results, mechanical ventilation status, lowest daily absolute (ABS) lymphocyte, and highest daily ABS neutrophil counts were abstracted for the first three weeks (21 days) after injury or until discharged. NLR was calculated using each daily ABS neutrophil and ABS lymphocyte count. Positive infection status was defined based on quantitative culture results of >50,000 colony-forming units (CFU) per milliliter, or moderate to heavy for qualitative culture results in all specimen types, except for cerebrospinal fluid (CSF) and blood, in which any presence was denoted as positive culture [42].

Among the initial injury measures mentioned above, the GCS score was used, which is a well-established assessment of neurological injury based on an individual’s verbal, motor, and eye responses [43]. ISS scores provide a comprehensive assessment of polytrauma injury severity to multiple anatomical regions by summing the AIS scores for the three most injured body regions [44].

### 2.3. Long-Term Global Outcome Assessment

The Glasgow Outcome Scale (GOS) was used to assess global recovery of participants at six months following their injury. On a scale from 1 to 5, GOS scores correspond to the following: (1) dead, (2) vegetative state, (3) severe disability, (4) moderate disability, and (5) good recovery [45]. For analysis purposes, GOS scores were dichotomized for two analysis groups as follows: Full cohort: unfavorable (GOS score = 1, 2, 3) and favorable (GOS score = 4, 5) outcomes; acute-care-survivors cohort: unfavorable (GOS score = 2, 3) and favorable (GOS score = 4, 5) outcomes.

### 2.4. Statistical Analyses

Statistical analyses were conducted in SAS Version 9.4 (SAS Institute Inc., Cary, NC, USA) and R, version 3.6.2, Vienna, Austria [46] in RStudio, version 1.3.959, Vienna, Austria [47]. Descriptive statistics including mean, median, standard error of the mean, and interquartile range (IQR) were used to describe continuous variables. Frequency measures and percentages were used for categorical variables. Non-parametric Mann Whitney or Kruskal Wallis U tests were conducted for continuous variables, and chi-square tests were used for categorical variables. All tests were two-tailed with a significance level set at α = 0.05.

### 2.5. Group-Based Trajectory Analysis 

Group-based trajectory (TRAJ) analysis can be used to model commonalities in the temporal dynamics of a particular dependent variable to identify distinct trajectory groupings. This methodology quantifies not only absolute levels of the dependent variable, but also temporal dynamics. In this study, both NLR trajectory group memberships were formulated using the SAS Macro *PROC TRAJ* [48], similar to our previous publications [49,50]. A log transformation was applied to daily NLR, daily absolute (ABS) Neutrophils, and ABS Lymphocytes datasets to fit a censored normal (CNORM) distribution for trajectory modeling. Bayesian Information Criterion (BIC) and average subject-specific posterior probabilities were used as model diagnostics for between-model comparisons (optimal number of trajectory groups and polynomial orders) and to determine the probability that individuals belong to the trajectory group assignment designated by the algorithm. Clinical judgement, along with model fit statistics (e.g., BIC) guided the formation of TRAJ groupings, which were used to explore associations with time to infection and associations with hospital resource utilization and global outcome.

### 2.6. Survival Analysis

Kaplan–Meier curves were used to determine if the estimates of the rate of non-infection over time differed between the NLR, ABS Neutrophils, and ABS Lymphocytes TRAJ groups. Survival analysis was used to assess “time to event” for each individual from a defined starting point [51]. In this study, “survival time” was defined as time elapsed from TBI occurrence to occurrence of first infection. This methodology determined the proportion of individuals without infection in each TRAJ group over the three-week monitoring period. This estimate provided the probability of no infection before time t and was used to compare the difference of this proportion between groups. This analysis required an assigned serial time frame (three weeks post-TBI), as well as infection status at each serial timepoint, and assigned group membership (TRAJ assignment) for all individuals in the cohort. Censored observations are accounted for in cases of death or acute care hospital discharge.

### 2.7. Changepoint Analysis and a Two-Part Model

To examine the temporal impact of NLR on infection status and vice versa, we explored the optimal changepoint for NLR TRAJ groups over time using the changepoint package in R [52]. Using this approach, we modeled relationships between time-to-infection and NLR over two independent time intervals. The optimal changepoint was defined as the day before and after which the statistical properties of the mean NLR sequence over time differed. The algorithm searches for the changepoint at which the two-part likelihood of the observed sequence was maximum. In the first interval, we generated a time-to-infection model from day zero until the optimal point of change in the mean trend of NLR using Cox regression and NLR TRAJ group membership while adjusting for age, sex, and GCS score as independent variables. This analysis was repeated for lymphocyte and neutrophil separately (instead of using NLR) for the first interval. To assess the reverse impact of infection on NLR, we also employed a mixed effects regression model using the log-transformed NLR (to account for skewness), time-varying infection status, and other covariates for this time epoch. In the second time interval, we generated another mixed effects regression model using the log-transformed NLR from the changepoint to day 20, using time-varying infection status while adjusting other independent variables. For the mixed effects regression models, we chose an autoregressive (order 1) covariance structure among the log transformed NLR values over time. The clinical and biological assumptions for these models were that NLR impacts the time to first infection until the changepoint, at which point afterward infection impacts NLR levels thereafter. 

### 2.8. Bivariate Analyses, Univariate and Multivariable Regressions of Hospital Resource Utilization, and Global Outcome 

Bivariate analyses (Mann Whitney and chi-square tests, as applicable) were used to compare demographic and clinical variables by NLR TRAJ group membership and infection status. Hospital resource utilization variables, including LOS and time on mechanical ventilation, and dichotomized GOS score were the outcome metrics for this study. Univariate linear regressions for each outcome variable were conducted using demographic and clinical variables including age, sex, GCS score (best in 24 h), ISS score, CT burden, infection status, and NLR TRAJ as covariates to inform variable selection for multivariable regression models using full and Acute-survivors-only cohorts. Hospital LOS and days on mechanical ventilation were modeled using linear regressions, and unfavorable (vs. favorable) global outcome (GOS) was modeled using a binary logistic regression. Variables associated with the dependent variable at a *p*-value < 0.1 in univariate regression analyses were incorporated into the multivariable models. Multicollinearity was inspected using the variance inflation factor (VIF) for each model, and a threshold VIF <10 was considered for each variable to reflect minimal multicollinearity effects [53]. For the binary GOS models for the full and Acute-survivors-only cohorts, receiver operating characteristic curves (ROC) and corresponding areas under the curves (AUC) were compared using DeLong’s test [54].

## 3. Results

### 3.1. Full and Acute-Survivors-Only Cohort Descriptions with Demographic, Clinical, and Hospital Resource Factors

Descriptions of both cohorts are provided in Table 1. In the full cohort, most patients were male (78.1%) and white (90.3%). Motor vehicle collision (MVC) was the most prevalent injury etiology (50.5%). The majority also presented with both intra-axial and extra-axial lesions (84.6% and 89.7%, respectively), had a median CT burden of three lesion types, acquired infection during their hospital stay (62.2%), and required mechanical ventilation (92.9%). In the acute-care-survivors only cohort, the majority were also male (78.3%) and white (89.7%). MVC was the most prevalent injury etiology (52.6%). The majority of those who survived three weeks post-injury also presented with both intra-axial and extra-axial lesions (83.9% and 89.7%, respectively), had a median CT burden of four lesion types, acquired infection during their hospital stay (64.6%), and required mechanical ventilation (92.0%).

### 3.2. Trajectory Groups and Corresponding Cell Counts

TRAJ analysis was used to delineate two trajectory groups for NLR: a *low* group consisting of 129 individuals (65.8%) and a *high* group consisting of 67 individuals (34.2%). The optimal fit for each NLR TRAJ group was a linear model. Adequate modeling metrics were met, as the average posterior probabilities for the *low* and *high* TRAJ groups were 0.9433 and 0.9424, respectively. Figure 2A shows a linear decrease in NLR over the first three weeks for both TRAJ groups. Interestingly, both groups start and end with significantly different NLR across the time course (*p* < 0.05 all comparisons).

Group-based trajectory analysis was also used to delineate two trajectory groups, a *low* TRAJ group and a *high* TRAJ group, for ABS lymphocytes and ABS neutrophils (Figure 3). For Figure 3 and Figure 2B,C, the dashed lines represent the normal clinical range as defined by UPMC clinical laboratory for ABS lymphocytes: 0.8–3.65 10^9^/L and ABS neutrophils: 2.24–7.68 10^9^/L. The optimal fits for both ABS lymphocyte TRAJ groups were a polynomial order of four, while the optimal fits for both ABS neutrophil TRAJ groups were also a polynomial order of four. Posterior probabilities were >0.9 for both TRAJ groups generated for ABS lymphocytes and ABS neutrophils. Concordance between NLR TRAJ vs. ABS lymphocyte TRAJ was significant (χ^2^ = 39.80, *p* < 0.0001), as was concordance between NLR TRAJ vs. ABS neutrophil TRAJ (χ^2^ = 37.02, *p* < 0.0001). However, concordance between ABS lymphocyte and ABS neutrophil TRAJ group membership was highly non-significant (χ^2^ = 0.0083, *p* = 0.927). The *low* ABS lymphocyte TRAJ group demonstrated clinical lymphopenia between days one and six post-injury, whereas the *high* ABS lymphocyte TRAJ group remained in the normal clinical range (Figure 3A). High ABS neutrophil TRAJ group members were, for the duration of the three-week hospital time course, clinically neutrophilic (Figure 3B). In contrast, the low ABS neutrophil TRAJ group started at an abnormally high level within the first two days post-injury and declined into the normal range for the remainder of the hospital stay except for a brief period of mild neutrophilia during days 9–13 (Figure 3B).

To further demonstrate these relationships, Figure 2B,C shows mean ABS neutrophils and ABS lymphocytes by NLR TRAJ group membership. As in Figure 3, mean neutrophil counts for the high NLR TRAJ group are consistently above the normal ABS neutrophils range, while mean neutrophil counts for the low NLR TRAJ fall to within the normal range during days 3–7, and again during days 15–20. Interestingly, both the high and low NLR TRAJ groups demonstrate similar temporal dynamics for mean neutrophil counts, such that each group demonstrated an early decline over the first five days, a rise during the second week, and a decline again during the third week. ABS lymphocytes drop precipitously over the first two to three days post-injury, but they remain in the normal clinical range for much of the three-week time course except for the *high* NLR TRAJ group during days 2–5 post-TBI, when first infection rates are high, and many in this group meet clinical criteria for lymphopenia. Resembling ABS neutrophil temporal profiles, ABS lymphocyte profiles for each NLR TRAJ group showed similar temporal dynamics to one another.

### 3.3. NLR TRAJ Associations with Demographic, Clinical, and Hospital Resource Factors

Bivariate associations between demographic and clinical measures were assessed in relationship to NLR TRAJ group membership (Table 2). Individuals in the high NLR TRAJ group tended to exhibit higher ISS scores. Those in the high NLR TRAJ group required more days of mechanical ventilation and had longer hospital LOS than the low NLR TRAJ group. Importantly, those in the high NLR TRAJ were more likely to have a nosocomial infection (High: 76.12% vs. Low: 55.04%; *p* = 0.006). Age, GCS score, sex, race, MOI, CT injury presentation and burden, and ventilator use did not significantly differ by NLR TRAJ group membership.

### 3.4. Post-TBI Infection Prevalence and Associations with Demographic, Clinical, and Hospital Resource Factors

Age, injury severity, and hospital resource utilization (time on mechanical ventilation and hospital LOS) were evaluated using bivariate analyses for associations with acute care infection status to survey the impact of injury severity on infection and of infection status on acute TBI clinical course. Table 3 reports that younger age (*p* = 0.003), more severe neurological injury based on GCS scores (*p* = 0.002), and CT burden (*p* = 0.007) were associated with infections. Those who acquired an infection during acute care also spent more days on mechanical ventilation and had a longer hospital LOS (*p* < 0.0001 both comparisons). We also characterized the sources of infections along with their relative prevalence, which can be found in Table A1 in the Appendix A.

### 3.5. Longitudinal Associations between Infection Acquisition and NLR

Figure 4A depicts density plots of new primary infections by day along with concur-rent NLR levels grouped by infection status. Here, daily NLR levels by infection status are overlaid by a frequency distribution of daily new infections reported. Days 3–6 had the highest numbers of new infections reported, although a minority still experienced their first infection during the end of the second week and throughout the third week post-TBI. Significant differences in NLR levels by infection status were noted on days 5–13 and 16–20, (*p* < 0.05 all comparisons). Visually, it appears that the highest rates of primary infections occur around day four, after which point, NLR levels start to separate by infection status. Interestingly, the infection positive group had higher NLR levels compared to the non-infected group. Figure 4B,C shows ABS lymphocytes and ABS neutrophils by infection status overlaid on the frequency distribution of new primary infections. Between days two and six, lymphocyte levels are slightly lower in the infected group compared to the group without infection (Figure 4B). Interestingly, Figure 4C shows a substantial separation in neutrophil levels between the no infection and infection sub-groups beyond day four and during the highest daily rates of new infections. ABS lymphocyte counts were lower in the infection group compared to the no infection group on days 5, 10–12, 16, and 19 (*p* < 0.05 all comparisons) (Figure 4B). ABS neutrophil counts were higher in the infection group compared to the no infection group on days 6, 8, 9, 15–17, and 20 (*p* < 0.05 all comparisons) (Figure 4C).

### 3.6. Kaplan–Meier: Time to First Infection

Kaplan–Meier estimates suggest the time to first infection was significantly earlier for a larger proportion of the *high* NLR TRAJ compared to the *low* NLR TRAJ (*p* = 0.003) group as reflected by the greater decrease in proportion of individuals without infection in the *high* NLR TRAJ (red) over time when compared to the *low* NLR TRAJ (blue) group (Figure 5A). Consistent with lower daily infection rates, noted in Figure 4, the proportion of new individuals acquiring infections declined among individuals in both TRAJ groups after day 10 post-TBI. Kaplan–Meier estimates of time to first infection were also generated by ABS neutrophil and ABS lymphocyte TRAJ groups. The proportion of individuals without infection was lower over time in the *low* ABS lymphocyte TRAJ group (*p* = 0.01) (Figure 5B). In contrast, there were no significant differences in time to infection between ABS neutrophil TRAJ groups (*p* = 0.20) (Figure 5C). Given the high degree of concordance between both ABS neutrophil and ABS lymphocyte TRAJ groups to NLR TRAJ groups, these results suggest that ABS lymphocyte counts, not ABS neutrophils, are likely responsible for the NLR TRAJ differences in time to infection, noted in Figure 5A.

### 3.7. NLR vs. Infection Changepoint Analysis: A Two-Part Model

We explored the optimal changepoint for NLR TRAJ groups over time and generated regression models for time-to-infection and NLR over two distinct time intervals. Our changepoint analysis of the mean NLR trend over time estimated that the distribution of NLR was different before and after day five (Figure 6). This finding was also supported by Figure 4B,C, wherein both ABS lymphocytes and ABS neutrophils began to rise again by days five and six, respectively, after an initial decline.

Thus, time to first infection was modeled during the 0–5 day period (Table 4). To minimize survivor bias within the analysis, participants who died within the study period before getting an infection were excluded from analyses. Participants in the *high* NLR TRAJ had 77.7% higher hazards of infection over the first six days after TBI (*p* = 0.041), adjusting for age, sex, GCS score, ISS score, and CT burden (Table 4A). The GCS score also independently predicted the time to first infection, wherein each unit increase in the GCS score was associated with an 11.5% decrease in the hazards of infection during this time period (*p* = 0.020). Similar to the Kaplan–Meier analysis above, participants in the *low* ABS lymphocyte TRAJ had 68.4% higher hazards of infection compared to the *high* ABS lymphocyte group (Table 4B), suggesting a causal role for lymphopenia regarding infection. There was no association between ABS neutrophil levels and time to first infection (Table 4C), suggesting neutrophilia is a consequence, not a cause, of infection.

As a second component of the changepoint analysis, a mixed effects regression model was performed for the days 0–5 and 6–20 log NLR values using an autoregressive (order 1) covariance structure (Table 5). During days 0–5, the log NLR was 0.1321 units higher (*p* = 0.072), and during days 6–20, the log NLR was 2.0237 units higher (*p* < 0.0001) among individuals with infection over time when adjusting for age, sex, GCS score, ISS score, CT burden, and days post-TBI. Log NLR decreased by 0.047 units over each day for the 0–5 days period (*p* = 0.005) and increased 0.27 over the 6–20 days period (*p* < 0.0001), independent of other variables. Log NLR also increased by 0.007 units over 0–5 days (*p* = 0.017) for each unit increase in ISS score. For a unit increase in CT burden, log NLR increased by 0.048 units over 0–5 days (*p* = 0.034), but the increase was not significant over 6–20 days (*p* = 0.172). Conversely, a unit increase in GCS score did not significantly contribute to change in log NLR over 0–5 days (*p* = 0.696), but significantly decreased log NLR by 0.022 units over 6–20 days (*p* = 0.022). While age did not play a significant role in NLR levels over day 0–5, younger individuals did have significantly lower NLR levels over 6–20 days (*p* < 0.0001).

### 3.8. Models of Hospital Resource Utilization

Due to the multiple interrelated factors presented that contribute to hospital resource utilization (LOS and time on ventilation) and, in turn, shape the ongoing recovery course, we utilized linear regression to produce models that covary for these contributing factors. Before performing multivariable models, we performed individual univariate regressions to assess individual covariate associations with LOS and ventilator days, which can be found in the appendix (Table A2 and Table A3). We included significant covariates (Table A2 and Table A3) into their respective multivariable models to obtain the adjusted effects for the hospital resource utilization analysis (Table 6). All VIFs were <2, indicating no significant multicollinearity among predictor variables.

After adjusting for covariates, factors that significantly increased LOS (*p* < 0.05) in both the full cohort and Acute-survivors-only Cohort included higher ISS score, higher CT burden, *high* NLR TRAJ group membership, and infections. Lower GCS score was significantly associated with higher LOS only in the Acute-survivors-only Cohort (Table 6A). Factors that significantly increased time of mechanical ventilation (*p* < 0.05) in both the full cohort and Acute-survivors-only Cohort were lower GCS score, higher CT burden, *high* NLR TRAJ group membership, and infections (Table 6B). In both models, infections had the largest association with hospital LOS and mechanical ventilation days.

### 3.9. NLR and Infection Impact on Long-Term Global Outcome

Before assessing the association between NLR TRAJ and infection with six-month global outcome in a multivariable model, individual univariate regressions were performed with clinical and demographic covariates, which can be found in Table A4. The results of the multivariable models are presented in Table 7 for both cohorts. After adjusting for covariates, factors that significantly increased the odds of an unfavorable outcome at six months (*p* < 0.05) in both cohorts were older age, lower GCS score, and higher CT burden (only marginally significant in the full cohort). In the full cohort, a unit increase in age increased the likelihood of an unfavorable outcome by 4.2% (OR = 1.042, *p* = 0.002). A unit increase in GCS score decreased the likelihood of an unfavorable outcome by 29.4% (OR = 0.706, *p* < 0.001). Participants in the *high* NLR TRAJ were 2.18 times more likely to have an unfavorable outcome compared to those in the *low* NLR TRAJ (OR = 2.18, *p* = 0.046). Higher CT burden showed a marginally significant association with the likelihood of an unfavorable outcome (OR = 1.29, *p* = 0.056). Among the acute-survivors-only cohort, the associations of the covariates with the unfavorable outcome were similar. Older age (OR = 1.029, *p* = 0.048), lower GCS score (OR = 0.77, *p* = 0.001), and higher CT burden (OR = 1.318, *p* = 0.047) were positively associated with an increased likelihood of an unfavorable outcome. *High* NLR TRAJ group membership was also marginally associated with a higher likelihood of having an unfavorable outcome (OR = 1.988, *p* = 0.087). VIFs for each of the included variables were <2 in both cohorts, suggesting no multicollinearity. ROC curves for the multivariable models using both cohorts are presented in Figure 7. AUC for the model using the full cohort was 0.81, whereas AUC for the model using the acute-survivors-only cohort was 0.77. DeLong’s test did not detect any significant difference between these two ROC curves (*p* = 0.474).

## 4. Discussion

### 4.1. Early Lymphopenia Drives Initial High NLR Levels Leading to Nosocomial Infections

For the past two decades, derangements in immune responses have been explored in the context of trauma and have focused on (1) systemic inflammatory response syndrome (SIRS), which is known for a predominance of proinflammatory cytokines [55], and (2) compensatory anti-inflammatory response syndrome (CARS), a frequently deleterious state of immunosuppression that often leaves patients vulnerable to infections [56,57,58,59]. These two immune syndromes can occur concurrently, or perhaps as a biphasic response [56,58]. CARS downregulates pro-inflammatory cytokines such as IL-1β and TNFα, both of which are responsible for the expansion and activation of lymphocytes [55,60]. CARS also results in increased anti-inflammatory cytokines which inhibit TNFα expression [57,61], limit T-cell activation and expansion, and reduce effector function [62], leaving lymphocytes with impaired cytotoxic and recognition capacities to fight nosocomial pathogens [63,64,65,66]. While TNFα secretion is a part of the hepatic acute phase response after TBI [67], IL-10 expression post-TBI impairs T-cell immunity, which decreases the ability to fight infection early after TBI [62,68].

Interestingly, TNFα expression can also initiate lymphocytic apoptosis via a complex interplay between FAS receptor and TNF receptor (TNFR)-II signaling, which we speculate may have a role in early lymphopenia and immunosuppression post-TBI [69,70]. Previous studies have found a rapid onset and clinically significant degree of lymphocyte apoptosis early on in post-trauma associated with septic shock and mortality [63,64]. Similarly, our study found that *high* NLR TRAJ group membership reflected an immediate decline in ABS lymphocytes, and for many in the *high* NLR TRAJ group, clinical lymphopenia occurred within days 2–5 post-TBI (Figure 2B). Thus, lymphopenia appears to drive subsequent development of infection (Table 4B), a finding that is also supported by trauma population studies, suggesting that early lymphopenia facilitates increased infection rates and longer hospital LOS [64,71]. Collectively, this evidence suggests that therapies aimed at preserving lymphocytes, either by reduction in lymphocyte anergy or apoptosis, or by supporting pro-lymphocyte factors, may represent a potential therapeutic approach for infection prevention in TBI. Since the activation of cognate receptors, namely TNFR-I (p55) and–II (p75), differentiate the role of TNFα in immune cell regulation [72,73], immuno-regulatory therapies often take the structural form of decoy receptors. Etanercept, for example, is a sTNFR-II decoy receptor that inhibits reverse (also known as outside-to-inside) signaling known to drive activated lymphocyte apoptosis and upregulate anti-inflammatory processes [73]. 

### 4.2. Infection and Subsequent Neutrophilia-Driven Increase in NLR

Early after TBI, damaged brain endothelium, astrocytes, and microglia upregulate pro-inflammatory cytokine production [74,75,76], facilitating adhesion molecule production and subsequent diapedesis of neutrophils through the blood brain barrier (BBB) into the brain parenchyma [77,78]. Once in brain parenchyma, neutrophils contribute to secondary injury cascades via multiple mechanisms that perpetuate BBB damage [79,80,81,82]. With concurrent infections, stem cell cytokines and other stimulatory factors increase circulating neutrophils [83] recruited to infected tissues where inflammatory pathways perpetuate ongoing neutrophil production [84,85], a phenomenon that could also indirectly exacerbate neutrophil-associated neuroinflammation. We found that patients having nosocomial infection within the first week of their hospital stay later had higher NLR levels that were driven by the increase in circulating neutrophils (Figure 4, Table 5B). Neutrophilia impacts on BBB damage, cerebral edema, subsequent neuroinflammation, and critical illness-associated SIRS may contribute to these findings (23 days versus 17 days in the *low* NLR TRAJ group).

### 4.3. Covariate Effects on NLR and Infection

We also considered the effects of injury severity, both neurological and non-neurological, on infection status and NLR. To help interpret these inter-relationships, we provide a conceptual summary of clinical variable associations by time window in Figure A1, which depicts a summary of the covariate associations derived from the changepoint and supplementary analyses. Lower GCS scores were associated with infections during both time epochs, likely due to the neurological need for mechanical ventilation with severe TBI and inherent risk of ventilator-associated pneumonia [11,12]. When evaluating NLR by time epoch, higher primary injury burden (as represented by CT burden) was associated with higher day 0–5 NLR profiles, while worse GCS scores had an inverse relationship with NLR profiles in days 6–20, a finding which may reflect the degree of secondary neurological injury that contributes to ongoing immune dysfunction. While the GCS score was associated with days on mechanical ventilation throughout the time-course, additional analyses showed that both those with early infection (day 0–5) and those with late infection (day 6–20) had increased ventilation days compared to those without infection, corroborating the hypothesis that both neurologic injury severity and infection contribute to days on ventilation over the time course evaluated. The ISS score was associated with NLR during days 0–5 post-injury, suggesting that the concomitant injury complex may also contribute to ongoing immune dysfunction; these injuries may also be a source of primary infection that perpetuates elevated NLR.

### 4.4. Applications and Utility of NLR in Traumatic Brain Injury

NLR has been explored as a prognostic indicator for mortality, outcome, and infection acquisition in many other fields of study; however, its implications for TBI are only now emerging. One of the factors contributing to the complexity of TBI research and treatment is the heterogeneity of the patient population with respect to injury severity and mechanism, personal biology, and comorbidity burden, along with variation in clinical care. This study highlights the contribution of nosocomial infection to TBI associated heterogeneity, for which NLR is both a contributor to and a readout of infection status. Our study utilized acute laboratory data to characterize how acute factors contribute to hospital utilization, complications, and outcomes. NLR potentially could be used as a marker to aid healthcare professionals in identifying higher infection-risk patients and in considering risk mitigation measures to alter the trajectory of care and resource utilization.

A common measure used to quickly assess TBI severity and prognosis is the GCS score, which is strongly associated with outcome [37]. The GCS has pitfalls, as its measurement is limited to when patients are sedated and/or mechanically ventilated. Therefore, additive markers that can be used in combination with the gold standard GCS may have more utility to discriminate subtle changes within the individual’s clinical course that may influence resource utilization and overall recovery. The potential for NLR to provide unique information about the hospital course (e.g., infection status, ventilator use, and hospital days) and subsequent outcome makes it an interesting candidate as it is an objective and routinely collected marker, unlike GCS which may be affected by clinical exam-related limitations [86]. Our study showed that, even after GCS score and other covariate adjustments, NLR TRAJ was still significantly associated with unfavorable outcome at six months (Table A4). Furthermore, the importance of the systemic immune response in TBI recovery further supports the potential use of NLR with TBI care.

### 4.5. Hospital Resource Utilization

One point of novelty with our study approach was the integration of infection status with the study of NLR effects on TBI outcomes. We observed a large effect of nosocomial infections on hospital resource utilization. Compared to those who did not acquire an infection during their acute care hospitalization, individuals with infection had a 13-day increase in hospital LOS and a 7-day increase on mechanical ventilation (Table A3). With more days of mechanical ventilation come increased costs, as previous literature points to mechanical ventilation being the greatest independent predictor of high intensive care costs [87]. While the breakdown between ICU and ward LOSs was not recorded, the incremental daily cost of ICU care for mechanical ventilation is estimated to be between $3000 and $5000 per individual [87,88]; thus, seven additional days of mechanical ventilation would be expected to increase the cost of care by $21,000–$35,000 per patient. Reid et al. estimated daily charges of US acute hospital stays for TBI to average $3000 [89]. Therefore, assuming an infection results in seven additional days of ICU care with mechanical ventilation and eight further days of inpatient hospital care (totaling $24,000), this would result in an overall increase in acute care cost of approximately $45,000–$59,000. This total does not take into account that the cost of care is rising annually [90], nor the cost of “unplanned” acute care hospital readmissions, which are common in TBI patients and are frequently attributable to infections [91]. A recent study at one medical center estimated the cost of unplanned acute care hospital readmissions for individuals with moderate-to-severe TBI to average between $2788 and $6702 [92], on top of the initial acute care costs. With more than 37.9 million Americans uninsured and with our cohorts’ age range disproportionately representing that uninsured population, these acute care costs place a large financial burden, especially on low-income TBI patients [93] and, ultimately, hospital systems. Using easily obtainable and low-cost clinical markers such as NLR to identify at-risk populations may help to reduce healthcare costs by providing targeted critical care and preventing HAIs that, when acquired, substantially increase these costs [94].

We observed strong relationships between younger age and high resource utilization, including longer hospital LOS and more mechanical ventilation days (Table A2 and Table A3), showing that younger patients had longer hospital courses. However, when analyzing with multivariable analysis (Table 6), the effect of age on resource utilization was attenuated, especially in survivors. We speculate this is secondary to younger patients sustaining mechanisms of injury that involve polytrauma complexes requiring multiple surgeries, whereas older patients are more likely to be victims of low energy mechanisms such as falls leading to more isolated brain trauma. We also observed that the GCS score significantly predicted both longer LOS and mechanical ventilation days in the acute-survivors-only cohort, but not in the full cohort. This finding also may be due to early mortality effects skewing these time-based measures.

### 4.6. Limitations

This study has several limitations. This was an observational study, so observed relationships between variables cannot be definitively established as causal or predictive. The retrospective abstraction of infection data was subject to medical record missingness and reliance on quantitative and qualitative culture reports as evidence of infections. It is possible prospective infection tracking would have yielded additional infection counts; however, our definitions of infection are in line with prior studies [42] and therefore should reflect clinically important infections. Future studies may benefit from clinical adjudications that also capture and qualify/quantify antibiotic use.

## 5. Conclusions

There is increasing recognition that the peripheral immune system plays a major role in secondary injury inflammatory cascades that accompany TBI, and that non-neurological injury [23,95,96], as well as infection [11,12,13] and critical illness [97,98], may perpetuate CNS damage, increase risks for secondary conditions and complications, and influence outcome. The goal of this study was to characterize how cellular immunity, represented by NLR, and early infection influence each other, hospital resource utilization, and global outcome. We found distinct TRAJ groups could be derived from daily NLR values that provide unique information about infection risk, hospital resource utilization, and global outcome. Specifically, individuals with *high* NLR trajectories were more likely to be clinically lymphopenic over days 0–5 post-injury and have elevated neutrophil levels over the three-week time course studied. *High* NLR trajectory group membership was also associated with increased infection rates and shorter time to infection, with lymphopenia having the largest influence on this relationship. Infection had a significant impact on neutrophilia, particularly over days 6–20 of the time-course observed. NLR trajectories and infection status were also associated with hospital resource utilization metrics, including longer hospital LOS and more days on mechanical ventilation. Finally, among the full cohort, *high* NLR was significantly associated with an unfavorable GOS score, a measure of global outcome, at six months post-injury. Interestingly, neurological injury severity had a significant impact on NLR profiles across the time span monitored, while polytrauma was associated with NLR 0–5 days post-injury. Together, these data suggest the importance of the early cellular immune system response to infection, hospital resource utilization, and long-term outcome after TBI, and the data suggest that NLR trajectory group membership may be a useful early biomarker in identifying individuals at risk for infection and poor outcome.

As mentioned previously, inflammatory mediators associated with both TBI and infections play key roles in neutrophil recruitment and proliferation. Future work should explore acute dynamic inflammation patterns and their associations with neutrophilia and lymphopenia, which could yield potential treatment targets for early intervention. In addition to understanding the biological basis of NLR trajectory dynamics after TBI, future research should also explore NLR relationships to acute non-neurological organ dysfunction, sepsis, and chronic TBI conditions with known links to inflammation such as epilepsy, cognitive dysfunction, and depression.

## Figures and Tables

**Figure 1 jcm-10-04365-f001:**
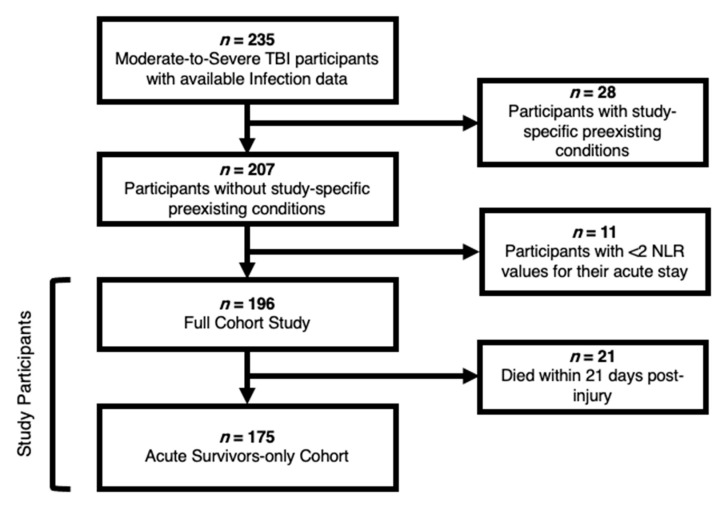
Study Cohort Selection Diagram.

**Figure 2 jcm-10-04365-f002:**
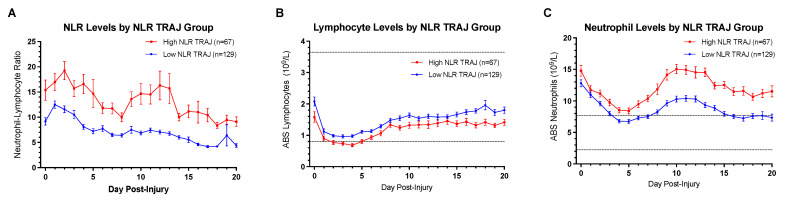
Daily Mean Levels by NLR TRAJ of (**A**) NLR, (**B**) ABS Lymphocyte, and (**C**) ABS Neutrophil Levels.

**Figure 3 jcm-10-04365-f003:**
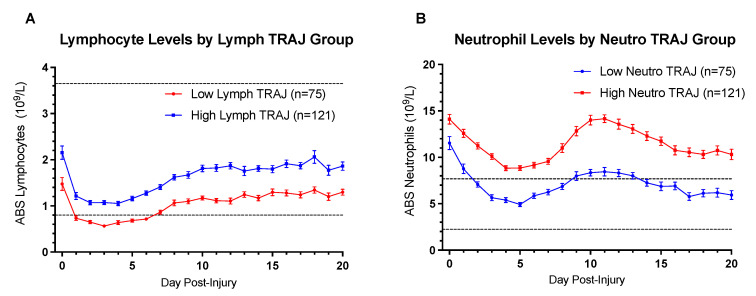
Daily Mean (**A**) ABS Lymphocyte Levels by Lymphocyte TRAJ group and (**B**) ABS Neutrophil Levels by Neutrophil TRAJ Group.

**Figure 4 jcm-10-04365-f004:**
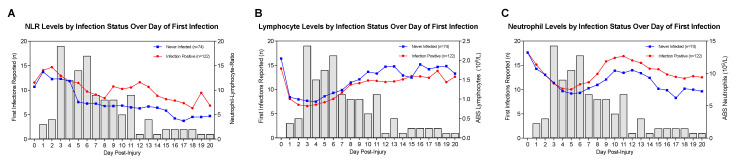
New Primary Infections Reported with Respect to (**A**) NLR, (**B**) ABS Lymphocytes, (**C**) ABS Neutrophils by Infection Status Over Three Weeks Post-TBI.

**Figure 5 jcm-10-04365-f005:**
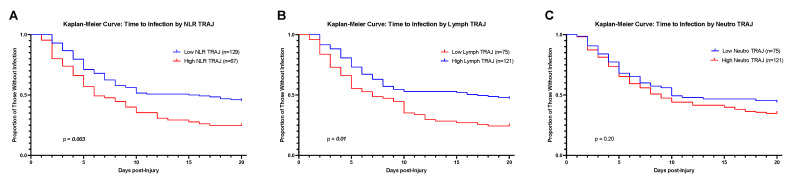
Kaplan–Meier Curves Depicting Time Until First Infection by (**A**) NLR Trajectory Group Membership (**B**) ABS Lymphocyte Trajectory Group Membership, and (**C**) ABS Neutrophil Trajectory Group Membership.

**Figure 6 jcm-10-04365-f006:**
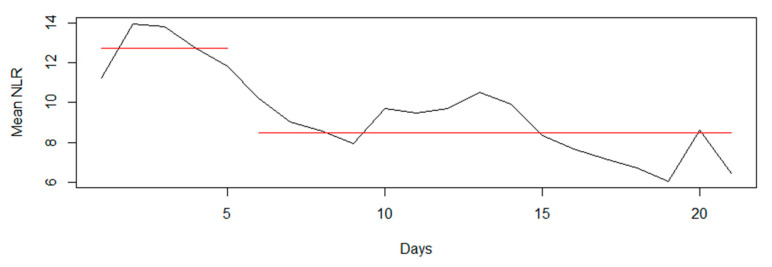
Changepoint Analysis of Mean NLR Trend Over Time.

**Figure 7 jcm-10-04365-f007:**
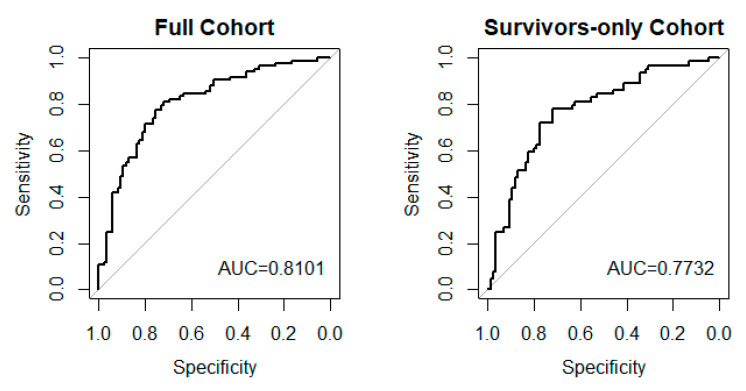
ROC Plots for Unfavorable GOS Using the Full and Acute-Survivors-Only Cohorts.

**Table 1 jcm-10-04365-t001:** Demographic and Clinical Characteristics for Full Cohort and Acute-Survivors-Only Cohort.

	Full Cohort		Acute Survivors-Only Cohort	
Variable	Median	IQR	Median	IQR
Age (Years)	31	(23–45)	29	(22–44)
GCS Score (Best in 24 h)	7	(6–10)	7	(6–10)
ISS Score	30	(25–38)	30	(25–38)
Length of Stay in Hospital (Days)	19	(10–27)	19	(12–29)
Mechanical Ventilation (Days)	10	(5–17)	11	(5–17)
	** *n* **	**%**	** *n* **	**%**
Sex (Male)	153	78.1%	137	78.3%
Race (White)	177	90.3%	157	89.7%
Mechanism of Injury	** *n* **	**%**	** *n* **	**%**
MVC	99	50.5%	90	52.6%
Motorcycle	36	18.4%	34	19.9%
Fall	37	18.9%	30	17.5%
Other	19	9.7%	17	9.9%
CT Injury Type/Complications	** *n* **	**%**	** *n* **	**%**
Intra-axial Hemorrhage	165	84.6%	146	83.9%
IVH	69	35.2%	58	33.3%
IPH	102	52.0%	91	52.3%
Contusion	128	65.3%	114	65.5%
Extra-axial Hemorrhage	175	89.7%	156	89.7%
SDH	129	65.8%	111	63.8%
SAH	145	74.0%	128	73.6%
EDH	41	20.9%	41	23.6%
DAI	31	15.8%	29	27.9%
Midline Shift	65	33.2%	56	32.2%
	**Median**	**IQR**	**Median**	**IQR**
CT Burden	3	(2–4)	4	(2–4)
	** *n* **	**%**	** *n* **	**%**
Infection Status (Positive)	122	62.2%	113	64.6%
Ventilator Use	182	92.9%	161	92.0%

**Table 2 jcm-10-04365-t002:** Demographic and Clinical Characteristics by NLR Trajectory Group.

	Low NLR TRAJ (*n* = 129)		High NLR TRAJ (*n* = 67)		
Variable	Median	IQR	Median	IQR	*p*-Value
Age (Years)	31	(23–46)	29	(22–44)	0.5084
GCS Score (Best in 24 h)	7	(6–10)	7	(6–10)	0.1481
ISS Score	30	(24–38)	33	(26–43)	0.0478
Length of Stay in Hospital (Days)	17	(9–23)	23	(15–33)	0.0013
Mechanical Ventilation (Days)	9	(4–14)	14	(8–20)	0.0015
	** *n* **	**%**	** *n* **	**%**	***p*-value**
Sex (Male)	101	78.3%	52	77.6%	0.9128
Race (White)	118	91.5%	59	88.1%	0.6089
Mechanism of Injury	** *n* **	**%**	** *n* **	**%**	***p*-value**
MVC	99	50.5%	90	52.6%	0.5560
Motorcycle	36	18.4%	34	19.9%	0.2046
Fall	37	18.9%	30	17.5%	0.2322
Other	19	9.7%	17	9.9%	0.4084
CT Injury Type/Complications	** *n* **	**%**	** *n* **	**%**	***p*-value**
Intra-axial Hemorrhage	107	83.6%	58	86.6%	0.7356
IVH	43	33.6%	26	38.8%	0.5719
IPH	66	51.6%	36	53.7%	0.8910
Contusion	82	64.1%	46	68.7%	0.6292
Extra-axial Hemorrhage	116	90.6%	59	88.1%	0.7548
SDH	85	66.4%	44	65.7%	1.0000
SAH	92	71.9%	53	79.1%	0.3547
EDH	30	23.4%	11	16.4%	0.3383
DAI	23	18.0%	8	11.9%	0.3749
Midline Shift	45	35.2%	20	29.9%	0.5575
	**Median**	**IQR**	**Median**	**IQR**	***p*-value**
CT Burden	3	(2–4)	3	(3–4)	0.7347
	** *n* **	**%**	** *n* **	**%**	***p*-value**
Infection Status (Positive)	71	55.0%	51	76.1%	0.0062
Ventilator Use	117	90.7%	65	97.0%	0.1813

**Table 3 jcm-10-04365-t003:** Demographic and Clinical Characteristics by Infection Status.

	Infected (*n* = 122)		Non-Infected (*n* = 74)		
Variable	Median	IQR	Median	IQR	*p*-Value
Age (Years)	29	(21–43)	40	(26–51)	0.0029
GCS Score (Best in 24 h)	7	(6–9)	9	(6–12)	0.0015
ISS Score	30	(26–38)	29	(20–36)	*0.0865*
Length of Stay in Hospital (Days)	22	(18–33)	9	(5–18)	<0.0001
Mechanical Ventilation (Days)	14	(9–18)	5	(2–9)	<0.0001
	** *n* **	**%**	** *n* **	**%**	***p*-value**
Sex (Male)	95	77.84%	58	78.38%	0.9334
Race (White)	110	90.16%	67	90.54%	0.2306
Mechanism of Injury	** *n* **	**%**	** *n* **	**%**	***p*-value**
MVC	65	54.17%	29	40.80%	0.603
Motorcycle	24	20.00%	12	16.90%	0.736
Fall	21	17.50%	16	22.54%	0.508
Other	10	8.33%	14	19.70%	0.736
CT Injury Type/Complications	** *n* **	**%**	** *n* **	**%**	***p*-value**
Intra-axial Hemorrhage	106	86.89%	59	80.82%	0.2561
IVH	52	42.62%	17	23.29%	0.0063
IPH	36	54.1%	66	49.32%	0.5175
Contusion	86	70.49%	42	57.53%	*0.0652*
Extra-axial Hemorrhage	110	90.16%	65	89.04%	0.8025
SDH	81	66.39%	48	65.75%	0.9272
SAH	96	78.69%	49	67.12%	*0.0735*
EDH	28	22.95%	13	17.81%	0.3937
DAI	24	19.67%	7	9.59%	*0.0624*
Midline Shift	41	33.61%	24	32.88%	0.9167
	**Median**	**IQR**	**Median**	**IQR**	***p*-value**
CT Burden	4	(3–4)	3	(2–4)	0.0065
	** *n* **	**%**	** *n* **	**%**	***p*-value**
Ventilator Use	121	99.18%	61	82.43%	<0.0001

**Table 4 jcm-10-04365-t004:** Changepoint Analysis Part 1: Time to Infection Model for First Five Days with (**A**) NLR TRAJ, (**B**) ABS Lymphocyte TRAJ, and (**C**) ABS Neutrophil TRAJ.

**(A) With NLR TRAJ**	**Beta**	**SE**	**χ^2^**	**Hazard Ratio**	** *p* ** **-Value**
Age	0.004	0.008	0.18	1.004	0.671
Sex (Male)	0.242	0.312	0.602	1.274	0.438
GCS Score	−0.144	0.049	8.802	0.866	0.003
ISS Score	−0.012	0.012	1.024	0.988	0.312
CT Burden	−0.104	0.083	1.567	0.901	0.212
NLR TRAJ (High)	0.575	0.247	5.410	1.777	0.02
**(B) With ABS Lymphocyte TRAJ**	**Beta**	**SE**	**χ^2^**	**Hazard Ratio**	** *p* ** **-Value**
Age	0.001	0.009	0.009	1.001	0.924
Sex (Male)	0.101	0.318	0.1	1.106	0.752
GCS Score	−0.139	0.049	8.136	0.870	0.004
ISS Score	−0.015	0.013	1.469	0.985	0.226
CT Burden	−0.096	0.081	1.4	0.908	0.237
Lymphocyte TRAJ (Low)	0.521	0.256	4.138	1.684	0.042
**(C) With ABS Neutrophil TRAJ**	**Beta**	**SE**	**χ^2^**	**Hazard Ratio**	** *p* ** **-Value**
Age	0.004	0.009	0.264	1.004	0.607
Sex (Male)	0.252	0.313	0.652	1.287	0.419
GCS Score	−0.152	0.049	9.419	0.859	0.002
ISS Score	−0.009	0.012	0.491	0.991	0.484
CT Burden	−0.114	0.084	1.821	0.892	0.177
Neutrophil TRAJ (High)	0.316	0.267	1.397	1.372	0.237

**Table 5 jcm-10-04365-t005:** Changepoint Analysis Part 2: Temporal log NLR profiles versus (**A**) Time-varying Infection Status (0–5 days) and (**B**) Time-varying Infection Status (6–20 days).

**(A) Time-Varying Infection Status (0–5 days)**	**Beta**	**SE**	**t-Value**	***p*-Value**
Infection Status (Positive)	0.132	0.073	1.80	*0.072*
Day	−0.045	0.016	−2.86	0.004
Age	−0.003	0.002	−1.31	0.189
Sex (Male)	0.107	0.081	1.31	0.191
GCS Score	0.005	0.012	0.39	0.696
ISS Score	0.007	0.003	2.39	0.017
CT Burden	0.049	0.023	2.12	0.034
**(B) Time-Varying Infection Status (6–20 days)**	**Beta**	**SE**	**t-Value**	***p*-Value**
Infection Status (Positive)	2.024	0.249	8.14	<0.0001
Day	0.274	0.058	4.73	<0.0001
Age	−0.046	0.005	−8.93	<0.0001
Sex (Male)	0.002	0.003	0.54	0.587
GCS Score	−0.222	0.097	−2.29	0.022
ISS Score	−0.012	0.015	−0.78	0.437
CT Burden	0.005	0.004	1.37	0.172

**Table 6 jcm-10-04365-t006:** Multivariable Linear Regression of (**A**) Hospital Length of Stay and (**B**) Days on Mechanical Ventilation.

**(A) Hospital Length of Stay**	**Full Cohort**			**Acute-Survivors-Only Cohort**		
**Variable**	**Beta**	**SE**	***p*-Value**	**Beta**	**SE**	***p*-Value**
Age (Years)	−0.166	0.064	0.01	−0.043	0.071	0.548
GCS Score	−0.304	0.337	0.368	−0.993	0.358	0.006
ISS Score	0.183	0.089	0.042	0.189	0.091	0.04
CT Burden	2.144	0.664	0.001	2.049	0.682	0.003
NLR TRAJ	3.942	2.001	0.05	6.217	2.099	0.004
Infection Status (Positive vs. Negative)	8.671	2.118	<0.001	6.078	2.304	0.009
**(B) Days on Mechanical Ventilation**	**Full Cohort**			**Acute-Survivors-Only Cohort**		
**Variable**	**Beta**	**SE**	***p*-Value**	**Beta**	**SE**	***p*-Value**
Age (Years)	−0.063	0.025	0.014	−0.032	0.029	0.263
GCS Score	−0.429	0.133	0.002	−0.64	0.146	<0.001
ISS Score	0.026	0.036	0.461	0.017	0.037	0.639
CT Burden	0.953	0.266	<0.001	0.833	0.277	0.003
NLR TRAJ	1.693	0.796	0.035	2.417	0.854	0.005
Infection Status (Positive vs. Negative)	4.738	0.836	<0.001	4.139	0.938	<0.001

**Table 7 jcm-10-04365-t007:** Multivariable Logistic Regression Models of Covariates to Six-month GOS.

	Full Cohort			Acute-Survivors-Only Cohort		
Variable	OR	96% CI	*p*-Value	OR	96% CI	*p*-Value
Age (Years)	1.042	(1.02, 1.07)	0.0020	1.029	(1.00, 1.06)	0.048
Sex (Men)	0.598	(0.23, 1.47)	0.269	0.533	(0.21, 1.34)	0.184
GCS Score	0.706	(0.60, 0.81)	<0.0001	0.77	(0.65, 0.89)	0.001
ISS Score	1.027	(0.99, 1.06)	0.124	1.026	(0.99, 1.06)	0.149
CT Burden	1.29	(1.00, 1.69)	*0.056*	1.318	(1.01, 1.75)	0.047
Infection Status (Positive)	0.714	(0.31, 1.61)	0.421	0.972	(0.40, 2.36)	0.95
NLR TRAJ (High)	2.18	(1.03, 4.77)	0.046	1.988	(0.91, 4.43)	0.087
**AUC**	0.81			0.77		

## Data Availability

The data presented in this study are available on request from the corresponding author. The data are not publicly available due to limitations with participant consent that restrict data sharing.

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
