# Peer review of "Neutrophil-to-Lymphocyte Ratios and Infections after Traumatic Brain Injury: Associations with Hospital Resource Utilization and Long-Term Outcome"

_jcm, 2021, doi:10.3390/jcm10194365_

Round 1

Reviewer 1 Report

TBI sequela are attributable to the brain injury as well as secondary neurologic injury.  Acute TBI patients may be more susceptible to in hospital infections due to a TBI induced in-balance of immune cells; infections are associated with poorer outcomes. This study found both high NLR and infections increased hospital LOS and vent days. In this population high NLR increased the odds of morbidity. High NLR trajectories were tied to lymphopenia over days 0-5 post-injury and associated with increased infection rates and shorter time to infection. Infection had a significant impact on neutrophilia, particularly over days 6-20. High NLR was also significantly associated with unfavorable outcomes six months after injury. Thus, early NLR levels may be useful in identifying individuals more at risk for in-hospital infections and poor outcomes.

Overall the paper is well written, but in the introduction especially I strongly encourage the authors to break up the many run on sentences and use the active voice when possible.

This paper is extremely dense! I appreciate the thoroughness and full description of methods and results, but suggest the authors think more critically about what to highlight to support the original hypothesis. The most critical results should be reported while the rest of the results should be presented in comprehensive appendix. Focusing on the primary data findings will increase the impact of the conclusion, which gets lost in all the extra analysis.

Abstract

  • Need a statement wrapping up findings, what is the primary conclusion?

Methods:

  • What was the reasoning for excluding previous head trauma? How was a previous head trauma determined for study exclusion, must it have been diagnosed and recorded in the EMR or did you ask the subjects? How accurate do you feel your methods was?
  • I would expect lines 134-142 to be in the results.
  • Typo in line 169, remove is

Results

  • Table 2: how are so many of the n values larger than the individuals in the group? e.g Low NLR TRAJ has 129 individuals, and 153 are male? It seems the data in table 1 is repeated here. This must be corrected.
  • Please include group ns in figure 5 as with others
  • Line 406: inde-pendently
  • This paper is dense, I would suggest including tables 5-15 and figure 7 to an appendix of find a way to condense this information.

Discussion

  • Limitations should be moved from conclusion to discussion

Reviewer 2 Report

This is well-written study exploring a novel concept of NLR ratio in TBI outcomes of hospital acquired infections, outcomes such as LOS, days on ventilation and GOS, and hospital resource utilization.

To my reading this draft clearly presents the rationale, methods, results, and discussion of the key themes. I recommend the following minor revisions (typographical):

  • Figure 1 (line 134) does not need to be bolded
  • in-text citations are not presented consistently (i.e. line 254 the citation is before the period, line 257 is after the period)
  • an additional period on line 261
  • Line 285 is missing the open bracket before "Figure 3"
  • line 295 - "demonstrates" should be "demonstrated"
  • throughout the paper there is unnecessary hyphenation that is likely a result of inputting the text into the template (line 291 be-tween, line 339 preva-lent, line 366 lym-phocyte, and many others)
  • line 442 - "Wilcoxon rank rum" = sum
  • alinement of Table 12 and Table 14 headings is not correct
